# SwimmerNET: Underwater 2D Swimmer Pose Estimation Exploiting Fully Convolutional Neural Networks

**DOI:** 10.3390/s23042364

**Published:** 2023-02-20

**Authors:** Nicola Giulietti, Alessia Caputo, Paolo Chiariotti, Paolo Castellini

**Affiliations:** 1Department of Mechanical Engineering, Politecnico di Milano, Via La Masa 1, 20156 Milan, Italy; 2Department of Industrial Engineering and Mathematical Science, Università Politecnica delle Marche, Via Brecce Bianche 12, 60131 Ancona, Italy

**Keywords:** vision-based underwater measurements, swimmers pose estimation, fully convolutional neural networks, athlete’s performance measurements

## Abstract

Professional swimming coaches make use of videos to evaluate their athletes’ performances. Specifically, the videos are manually analyzed in order to observe the movements of all parts of the swimmer’s body during the exercise and to give indications for improving swimming technique. This operation is time-consuming, laborious and error prone. In recent years, alternative technologies have been introduced in the literature, but they still have severe limitations that make their correct and effective use impossible. In fact, the currently available techniques based on image analysis only apply to certain swimming styles; moreover, they are strongly influenced by disturbing elements (i.e., the presence of bubbles, splashes and reflections), resulting in poor measurement accuracy. The use of wearable sensors (accelerometers or photoplethysmographic sensors) or optical markers, although they can guarantee high reliability and accuracy, disturb the performance of the athletes, who tend to dislike these solutions. In this work we introduce swimmerNET, a new marker-less 2D swimmer pose estimation approach based on the combined use of computer vision algorithms and fully convolutional neural networks. By using a single 8 Mpixel wide-angle camera, the proposed system is able to estimate the pose of a swimmer during exercise while guaranteeing adequate measurement accuracy. The method has been successfully tested on several athletes (i.e., different physical characteristics and different swimming technique), obtaining an average error and a standard deviation (worst case scenario for the dataset analyzed) of approximately 1 mm and 10 mm, respectively.

## 1. Introduction

Nowadays, the improvement of sports performance at a competitive level is based not only on the instructions of coaches, but also on the observation of videos in which athletes are recorded during training. In this way, an athlete’s technique can be observed in detail by both the coaches and the athlete in order to better identify potential errors in the execution of the exercises and to correct them. Specifically, in swimming, the athlete’s performance can be assessed by extracting information on stroke rate, leg kick frequency and body posture [1]. In some cases, sensors equipped with accelerometers are used to derive parameters related to the movement of the athlete; in other cases, photoplethysmographic sensors, such as smartwatches and chest-worn devices, are used for the physiological monitoring during sports [2]. However, these solutions are highly sensitive to motion artifacts, uncomfortable to wear during the performance and inconvenient from a fluid dynamics perspective. Moreover, the use of these sensors does not allow obtaining information about the swimmer’s pose, which is of paramount importance. To overcome these issues, optical markers are typically attached to the human body to accurately reconstruct motion. Nevertheless, some markers are expensive, bulky and tend to affect the athlete’s performance; and in some cases, it is not possible to detect them in water due to the refractive index, so marker-less methods are being used to extract human poses [3]. In order to immediately obtain information about the athlete’s pose, computer vision algorithms play an important role in the automatic video analysis [4]. In fact, image processing is used in order to save time and resources that, until now, have been used in manually determining the key poses [5,6]. When reconstructing a human model through computer vision, some approximations should be considered, since the human body is a non-rigid, flexible and complex system. For example, human limbs are addressed as rigid elements that are capable of movement and connected together at certain points, which represent articular joints [3]. There are numerous works in the literature for marker-less automatic pose estimation of athletes in various disciplines based on the combined use of computer vision and artificial neural network (ANN) models (i.e., Mediapipe, Convolutional Pose Machine, OpenPose, etc.). However, these models fail when applied in underwater contexts due to the presence of turbulence, bubbles and numerous other disturbing elements [7,8,9]. Several end-to-end models for human semantic parsing and pose estimation also exist in the literature (e.g., see [10,11], to name a few). These models ensure that the positions of key body points can be derived directly from the input image in a short time, without intermediate steps and without involving sophisticated post-processing algorithms. The effectiveness of these networks is also based on the availability of numerous databases for human semantic parsing (e.g., ATR, CCF, CIHP, LIP, Pascal and PPS, to name a few), with which they can be trained [12]. However, none of these are effective at recognizing the poses of swimmers in an underwater environment, and no annotated datasets of people underwater are publicly available. In recent years, several works have dealt with marker-less pose estimation of swimmers. Image quality is one of the necessary features for achieving good motion capture. For this purpose, cameras are totally or partially submerged in the pool, as needed. In particular, Cohen et al. [13], in their study, used a pair of cameras located on a trolley which was able to follow the swimmer during the performance. One camera was positioned above the waterline and the other one was submerged below. Greif et al. [14] used a stationary camera setup: two cameras were mounted to a rack so that both underwater and above-water views were obtained. Einfalt et al. [15] used a single stationary camera behind a glass pane, side-on to the athlete. This set-up made it possible to record the movement of the athlete both above and below the water’s surface. Regardless of the camera’s location within the pool, the problem of ensuring good image quality still remains crucial due to the in-water environment. In fact, the correct estimation of poses is affected by the presence of bubbles and water splashes due to the fast movement of the athlete’s arms and legs during the swimming activity [6]. In addition, reflection and refraction phenomena in water make the pool a hostile environment for camera shots. This is particularly challenging when trying to match underwater and overwater images framing the same event [4]. In addition, frequent swapping between the pose prediction of the right and left limbs makes trajectory reconstruction laborious. For this reason, optimization algorithms in the post-processing phase are absolutely necessary. Moreover, since the training of an ANN model requires manually annotated input masks, the expert user’s errors in identifying the precise location of the targets are to be taken into account. Zecha et al. [16] tried to improve pose estimation by making a partition of the graph that allows each target’s label to swap if its trajectory fits better with another target’s trajectory. However, their setup was in a laboratory environment, where the swimmer was constantly visible from a side view, below and above the free water’s surface, through a swimming glass channel. In real conditions, this configuration would require complex equipment to be able to follow athletes with the camera in real time. In addition, it would be difficult to make these captures during a swimming competition without affecting the athlete’s performance. Indeed, when targeting the measurement of athletes’ performances, it is of primary importance to identify target body parts accurately. During underwater filming, some body parts are very often partially hidden because the swimmer is framed laterally; moreover, the presence of turbulence and bubbles, combined with the hostile filming environment, may not guarantee that the athlete’s body as a whole will be captured at all instants. Therefore, the task of measuring athletes’ performances should not be approached as a standard human parsing or skeleton detection problem. Indeed, the focus should also be on ensuring uniformity and stability of results when assessing the position of a target body part. For example, if only the right hand and left foot are visible in a frame due to a particular movement of the athlete, or due to the presence of strong turbulence, one should not attempt to reconstruct the body in its entirety: priority should be given to the identification, with the greatest possible accuracy, of the target body parts that are clearly visible. It will then eventually be up to the coach, when post-processing the data, to interpret the data correctly and compensate for the non-detection of certain target body parts in certain frames. For this reason, this should be approached more as an object detection problem of the target body parts involved in analyzing the movement of swimming athletes. In fact, unlike the more typical approaches to skeleton pose estimation/human parsing problems, for the purposes of analyzing the technical gesture and returning performance indices, it is of fundamental importance not to reconstruct the individual frame, but rather the trends over time of the positions of the various targets, in order to then be able to derive information on reciprocal position, speed and acceleration. Furthermore, it is crucial to note that there is no need to work at high frame rates, because the analysis is done at a later stage than the swimming phase and is used as a gesture correction tool by the coach at an even later stage. Therefore, priority should not be given to computational time, but rather to the accuracy of identifying the target body parts. Based on these assumptions, in this work we introduce SwimmerNET, a novel method based on computer vision and fully convolutional neural networks (FCN) that makes it possible to estimate the poses of top-class swimmers by exploiting underwater images acquired in a real pool. The developed method aims at overcoming all the above mentioned problems associated with the underwater pose estimation of swimmers while ensuring high accuracy in locating target body parts. The developed method was tested in a setup consisting of a single fixed camera with an horizontal angular field of view (AFOV) of $63^{\circ }$, ensuring that multiple cycles of the swimming gesture are captured. Nevertheless, the method perfectly fits other set-ups, involving the use of either multiple fixed cameras or single/multiple non-fixed cameras, whose positions should be tracked and recorded, in such a way that the entire swimming lane can be covered in the analysis. Trajectories and anthropometric measurements of athletes can thus be used to help coaches in assessing their athletes’ performance parameters. The developed method has been successfully tested on professional athletes, and the results were compared with the manual annotations of an experienced swim trainer in order to evaluate the metrological performance of the developed method.

## 2. Material and Methods

Our experiments targeted the swimming performances of male and female top-class athletes in standard 25 m swimming pools. The selected swimmers had different technical characteristics: some focused more on sliding and some more on propulsion. Additionally, anthropometric and ergometric parameters of each athlete were measured to observe how much they affect performance.

### 2.1. Image Acquisition System

One single GoPro HERO 10 action camera was used for all acquisitions. This recording device was chosen to demonstrate the versatility of the approach developed. Underwater videos were recorded at a resolution of 3840 × 2160 pixel at 120 fps. A specific structure was made to submerge the camera underwater, allowing it to remain fixed at a height of 10 cm below the free water’s surface. The acquisition system was located on one side of the swimming pool, as shown in Figure 1. A side view is useful, as it provides a wide field of view and allows observing more swimming cycles while keeping the camera’s position stationary. The chosen camera’s horizontal AFOV ($63^{\circ }$) ensures a framed horizontal field of about 9 m at a distance of 6 m from a swimming lane. The camera was calibrated using targets of known size positioned underwater to reduce the effect of perspective distortion, resulting in a pixel-to-mm conversion factor of 1 pixel = 5 mm.

### 2.2. AI-Based Marker-Less Pose Estimation

#### 2.2.1. Model Architecture

The semantic segmentation of the body parts in each image was achieved by using multiple FCN-inspired architectures, which receive a single image as input and provide as output a segmentation mask (i.e., a binary image in which the value of 1 is assigned to the pixel belonging to the target body part and 0 to any other pixel). Various binary semantic segmentation architectures, chosen from the most modern ones available in the literature, were tested. Typically, the common architecture consists of two parts: a contraction path (encoder) that reduces spatial information and increases feature information, and an expansion path (decoder). The architecture can accept different model architectures as encoders. At first, it encodes the image by passing it through a CNN architecture and then performs up-sampling to obtain the output segmentation mask [17].

#### 2.2.2. Model Training

Based on the indications of the swimming coaches, 7 parts of the athlete’s body were selected to be tracked during the swimming session: ankles, knees, hip, wrists, elbows, shoulders and head. The targets that were then used for the final analysis were 13 in number, since, apart from the head, all those body parts exist in duplicate (Figure 2).

Each acquired video frame was labeled manually by an experienced swimming trainer through a graphic user interface specifically developed in Python. Only the visible target body parts were labeled. Those occluded by other body parts were not annotated. The resulting mask represented the ground truth for training the semantic segmentation model. The athlete’s whole body was also labeled in order to train the model to find the athlete’s body in each frame. In addition, based on the position of the body within the image, a region of interest (ROI) was selected, and the search field of the target body parts was restricted, so that the body’s reflection and any objects inside the pool did not affect the recognition of body-part targets. A binary semantic segmentation model was then trained for each defined target (i.e., 7 body parts and the athlete’s whole body, resulting in a total of 8 binary classification models being trained). Using a multiplicity of binary semantic segmentation models instead of a single multi-class semantic classification model ensured the robustness of each model. In fact, some targets were more often present than others in the framed scenes, generating a strong imbalance between classes (e.g., the left pelvis always ends up hidden if the video captures the athlete swimming freestyle in the swimming lane from left to right). The use of multiple binary semantic segmentation models allows the models to be trained solely on a single target, ensuring high accuracy for each one, at the expense of computational speed and resource consumption [18]. Data augmentation techniques (horizontal and vertical flipping, random displacement, random rotation, Gaussian noise and perspective transformation, just to mention a few) were used to increase training performance [19]. The Bayesian optimization approach, already used in Ref. [20], was exploited in order to identify the hyper-parameters providing the best results for the models. This technique has proven to be superior to classic random searching and grid searching [21]. The first step of this optimization method consists of defining, based on the problem, the hyper-parameters to be optimized in order to maximize the model’s metrics. The range of variability of each hyper-parameter is chosen according to the nature of the problem. The algorithm then performs the training of the model multiple times, after selecting a random set of hyper-parameters. Based on the values obtained in the former step, the Bayesian optimization process begins. At each stage, the algorithm evaluates the information passed on the model to select new optimal parameter values that increase the performance of the latest model, according to the method described by Agrawal [22]. The Jaccard index, also know as intersection over union (IoU), was used as a metric for evaluating the performances of the segmentation models (Equation (Equation 1)). This metric quantifies the overlap between the ground truth and the output prediction by measuring the ratio between the number of pixels shared by the ground truth and the predicted mask and the total number of pixels representing the two masks. This metric ranges from 0 (no overlap) to 1 (perfect overlap). We evaluated the model’s worst-case performance. The objective function to be optimized was defined as Equation (Equation 2), where *x* is the input image and *y* is the ground truth. The input parameters of the function that were made to vary in such a way as to maximize the IoU score were:Architecture *a*, which defines the type of architecture used for binary semantic segmentation. These were selected from the most recent ones available in the literature: Unet, Unet++, Linknet, PSPNet, DeepLabV3 and DeepLabV3+ [17].Batch size bs, which defines the number of training-data sub-samples that will be propagated through the network. The batch size was varied between 2 and 16.Learning rate lr defines how severely the model will change in response to the estimated IoU score each time the model’s weights are updated by the optimizer. The learning rate was varied between 0.0001 and 0.1.Optimizer *o*, which defines the algorithm exploited to increase the IoU score by modifying attributes of the neural network, such as weights and learning rate. The optimizer can be selected among SGD, Adam, RMSprop, Adadelta and Adagrad [23].Backbone bb, which is the inner CNN architecture of the encoder path. The optimizer can be selected among efficientnet-b7, efficientnet-b6, efficientnet-b5, inceptionv4, vgg19, vgg16 and mobilenet.
(1)$$IoU=\frac{Ground\;\;Truth\;\cap \;Predicted}{Ground\;\;Truth\;\cup \;Predicted}$$
(2)IoU=IoU(x,y,a,bs,lr,o,bb)

Everything was implemented via custom Python code using the Segmentation Models library [24], for handling the various architectures, loss functions and backbones, and the Bayesian Optimisation library [25], for handling the Bayesian optimizer.

#### 2.2.3. Pose Optimization Algorithm

The models developed in Section 2.2.2 made it possible to obtain 8 binary masks for each frame: a value equal to 1 was assigned to the pixel belonging to the target body part, and a value of 0 was given to any other pixel. In this way, the areas where the target body part was located were identified. The position of the target body part was identified as the pixel location (i,j) of the center of mass of the predicted area. Targets that involve two body parts (e.g., left and right wrist) provide two locations as output. The location of each target was then stored and identified through the expression $p_{n}\left(f\right)=p_{n}\left(i,j,f\right)$, where *n* is the body part’s ID (Figure 2) and i,j represent the row and column coordinates, in pixels, based on the center of mass for frame *f*. The image reference system was located at the top-left corner of each image, and the real-world coordinate reference system was chosen to have underwater depth data (*y* axis) reported as negative values. From the time sequence of each body-part target’s position, information related to the athlete’s performance can be extracted. Indeed, the trajectory of each target can be tracked. However, some interruptions in the trajectory can take place. These interruptions can be due to:the absence of targets during the over-water motion;the choice of a single viewpoint (side-view) of the camera, which prevents the identification of one side of the body;the swap between right and left targets, as the models developed do not distinguish between the two parts (e.g., with the exception of the head, all other body parts relate to left and right sides).

The first two issues are a direct consequence of the use of a single camera. This is not a true limit, as this choice just reduces the range of movements that can be analyzed. The third issue is much more disturbing and does not make an automatic interpretation of the trajectory possible. Indeed, as mentioned, the models described and developed in Section 2.2.2 are not able to distinguish between left and right parts of the body. Consequently, a specific algorithm was developed to overcome this issue. For each target body part, its associated model may return as output zero or more masks within the same frame. To handle any potential inconsistency, it is assumed that a maximum of two masks should be handled by the algorithm. This requires the definition of two set of mask locations, $p_{n,A}\left(f\right)$ (set A) and $p_{n,B}$, for each body part. However, different situations might occur. If no masks are returned for that particular body part, $p_{n,A}\left(f\right)$ and $p_{n,B}\left(f\right)$ are assigned the value NaN (not a number); this may occur in cases where the network cannot locate the target body part, or because the target body part is completely hidden and not present in the frame. In cases where just a single mask is returned as the model’s output, the algorithm assigns the location of the mask’s center of gravity to set A, and set B is assigned the value NaN. This may occur if the body part to be found does not include a right or left part (i.e., whole body or head), or because the model fails to identify a part, or if the missing part remains hidden in the current frame. Whenever two or more masks are found, these masks are sorted in descending order according to their levels of extension in pixels. The location of the mask showing the largest area is assigned to set A, and the location of the mask showing the second largest area is assigned to set B. In the rare event that more than two masks are identified, the small masks are considered as noise and not involved in the analysis. The labeling approach described so far clearly highlights that the two sets of mask locations, i.e., set A and B, cannot represent the left and right parts of the targets, so a solution must be found to rearrange the points so that set A contains only the right parts and set B the left parts, or vice versa; otherwise, the trajectories of the individual targets would be meaningless. The optimization algorithm developed to re-order this set of data works on velocity and acceleration data calculated on the masks locations and grounds on the general idea that swapped velocity and acceleration data values result in high and nonphysical values. Velocity and acceleration are computed following Equations (Equation 3) and (Equation 4), where i,j∈{A,B}; dt is defined as the reciprocal of the acquisition frame rate, $f^{\prime \prime }$ is the frame following that one/those ones where the target body part *n* is missing (i.e., the frame following a series of one or more NaNs) and $f^{\prime }$ is the frame preceding that one/those ones where the target body part *n* is missing (i.e., the frame preceding a series of one or more NaNs). Acceleration data are exploited only when a target body part is missing. In fact, velocity data for a frame containing NaN values will result in NaN values as well, thereby preventing their use. Acceleration data, specifically calculated between frames of different intervals, can solve the issue of being too sensitive to a lack of data.
(3)$$\begin{array}{c} \left\{\begin{array}{c} v_{ii}\left(f\right)=\frac{p_{n,i}\left(f\right)-p_{n,i}\left(f-1\right)}{dt}\\ v_{ij}\left(f\right)=\frac{p_{n,j}\left(f\right)-p_{n,i}\left(f-1\right)}{dt} \end{array}\right. \end{array}$$
(4)$$\begin{array}{c} \left\{\begin{array}{c} a_{ii}\left(f\right)=\frac{v_{i}\left(f^{\prime \prime }\right)-v_{i}\left(f^{\prime }\right)}{t\left(f^{\prime \prime }\right)-t\left(f^{\prime }\right)}\\ a_{ij}\left(f\right)=\frac{v_{j}\left(f^{\prime \prime }\right)-v_{i}\left(f^{\prime }\right)}{t\left(f^{\prime \prime }\right)-t\left(f^{\prime }\right)} \end{array}\right. \end{array}$$

In Figure 3, is it possible to observe examples of wrist trajectories before (a) and after (b) the application of the proposed optimization algorithm, for which the two sets of data were re-ordered by exploiting the proposed method. It is well evident that the proposed algorithm resolves the inconsistency in mixing the assignment of a body part to sets A and B, thereby making it possible to perform robust analyses of the tracked body parts.

#### 2.2.4. SwimmerNET Workflow

The complete procedural workflow we propose for marker-less pose estimation of swimmers can be described as follows. Once the video of the athlete is acquired, the algorithm starts by taking the first frame (f=0) as input. As anticipated in Section 2.2.2, the whole-body binary semantic segmentation model is applied to be able to identify the position of the athlete within the frame *f*. Based on the center of gravity of the mask (i.e., athlete’s center of gravity) that is returned as output from the model, a rectangular ROI is identified, with a fixed size of 1024 × 576 pixel. If the ROI falls outside of the image (i.e., the athlete is too close to a side edge of the frame), the frame is discarded and the next frame (f=f+1) is analyzed. These steps are critical to ensure that, from here on, only full, fixed-size ROIs with the swimmer always in their center are analyzed (Figure 4).

The 7 models developed in Section 2.2.2 for semantic segmentation of body parts are then applied to the ROIs. Each model returns as output a binary mask highlighting the area involving the target. The locations in pixels ($p_{n}\left(i,j,f\right)$) of the targeted body parts *n* are identified in the current frame *f*, as described in Section 2.2.3. For example, the binary mask identified by the semantic segmentation model for target ‘head’ (i.e., the swimmer’s head) and superimposed on the input image is shown in Figure 5.

At this point, for each correctly processed frame, a coordinate is assigned for each visible body-part target (Figure 6).

Once all targets in all acquired frames are identified, they are post-processed by the algorithms developed in Section 2.2.3 to reduce errors and distinguish left and right body parts. The correct poses of all targets are finally used for extracting the metrics needed to determine the athlete’s performance (mean velocity, frequency, fluctuations, force, thrust, etc.).

## 3. Results and Discussion

### 3.1. Swimmer’s Body Parts Semantic Segmentation Model Training

A total of 2021 frames, taken from 4 different videos recorded as described in Section 2.1, were collected for training the models. The videos involve an adult male athlete during freestyle swimming sessions in a single pool selected for testing. Frames were randomly selected from the videos in order to increase the variability of the data. It is important to note that the quality and type of movement of the athlete does not influence the training of the models, as each of them focuses on a single frame and not on their temporal sequences. The manually labeled masks (i.e., model’s ground truth) and the correspondent frames (i.e., model’s input) were randomly split into training, validation and test datasets according to the scheme 8:1:1 (i.e., training dataset length = 1615; validation dataset length = 404; test dataset length = 404). Training and validation dataset were used for training the eight semantic segmentation models. The models’ hyper-parameters were tuned following the Bayesian optimization technique described in Section 2.2.2 in order to maximize the IoU score. The models were trained on a machine equipped with 11th Gen Intel(R) Core(TM) i7-11700 @ 2.50GHz and 2 NVIDIA GPU RTX 3090 (e.g., total of 48 GB GPU memory). With this configuration, a single training session can be carried out in about four hours, considering a maximum number of 500 epochs. However, the time taken is typically considerably less, as the training is stopped early if the validation loss does not improve within five epochs. As shown in Table 1, all models achieved an IoU score of 0.999 using the DeepLabV3+ as the architecture, Adam as the optimizer and CNN efficientnet-b7 as the backbone. All models were cross-validated by the k-fold approach (five folds), obtaining a standard deviation of IoU score of less than 0.001 every time [26]. It is remarkable that, using different architectures, the performance of the model stayed the same, but with more modern architectures, the same result was achieved in fewer training epochs. In fact, on average, with the DeepLabV3 architecture, an IoU score of 0.999 was obtained in about 10 epochs. With Unet, Unet++, Linknet, PSPnet and DeepLabV3, 50, 42, 45, 40 and 22 epochs were needed, respectively.

### 3.2. Method Performance Analysis on New Videos

The SwimmerNET method, which is explained in Section 2.2.4, was then applied on three newly recorded videos (i.e., not used for model training), and the trajectories of all body-part targets were calculated. Consequently, each swimming cycle could be analyzed in detail by isolating all body-part targets. Each video was about 10 s long and recorded at 120 fps. A total of about 3600 different frames were processed by the algorithm. The new videos were recorded from different viewpoints, in two different pools and with different lighting conditions (Figure 7). Moreover, these new videos targeted two other athletes (different in gender and physique from the the athlete used for training the models) during swimming sessions involving different styles: dolphin (Figure 7b), freestyle (Figure 7c) and backstroke (Figure 7d). Moreover, frames extracted from the public Sports in the Wild (SVW) [27] repository were also analyzed. SVW is an archive of sports video data used for computer vision and machine learning research. The dataset contains various sports activities, such as basketball, football and swimming, recorded under realistic and challenging conditions, including cluttered and dynamic backgrounds, camera movement and player occlusions. The goal of SVW is to provide researchers with a challenging and diverse dataset for the evaluation and development of computer vision algorithms for analyzing and understanding sports. Within the dataset, there is also underwater footage of swimmers during freestyle (Figure 7e,f), and this was used to test the developed method. The dataset provides information regarding the position of the athlete(s) within the frame (i.e., the bounding box). This made it possible to effectively assess the performance of the proposed method. All target body parts were hand-labeled manually by an experienced user, as previously described in Section 2.2.2. Figure 7a also shows a video frame used for training the models, to highlight the differences between the training and testing videos.

The algorithm managed to process, on average, one frame every 0.8 s (it should be noted that the computation time necessarily depends on the hardware configuration used). Figure 8 shows two examples of trajectories of the right parts of the body during a swimming stroke: (Figure 8a) freestyle performed by a 1.90 m tall male athlete and (Figure 8b) dolphin style performed by a 1.80 tall male athlete.

The procedure for estimating the trajectory was the same for each target, but each target had its own specific features linked to visibility, speed of movement, shape and the variations in the same during movement (i.e., the wrist could bend while changing shape or rotate and be seen by the camera in a different way).

The performance index of the proposed measurement system denotes the ability to recognize and locate body-part targets in the various frames. The error in locating target position was defined as in Equation (Equation 5):(5)$$\left(e_{i},e_{j}\right)=\left(i_{gt}-i_{pm},j_{gt}-j_{pm}\right)$$
where $\left(i_{gt},j_{gt}\right)$ identifies the the reference target location (i.e., the ground truth defined by the experienced swimming athletic trainer) and $\left(i_{pm},j_{pm}\right)$ the target location calculated by SwimmerNET.

Figure 9 shows the bar chart with the percentage of frames in which the targets were not recognized, although they were present and visible within the image. A target is considered unrecognized when the model does not report its presence (i.e., the output binary mask is composed of all null values), or when the target is confused with another target or a disturbance object present in the framed scene. The percentage of frames in which the targets are not recognized is quite low, and moreover, the skipped frames are isolated and not synchronous between the various targets, allowing effective filtering and interpolation of the trajectories by applying the algorithm described in Section 2.2.3. It is also important to note that in frames where no target was present, the method never provided a false position, exhibiting fairly robust behavior.

Figure 10 shows how standard deviation of the absolute error in locating target position is affected by outlier data (i.e., target not recognized). In fact, after filtering out the outlier, significant improvement of the standard deviation can be observed. Once the outliers are removed, the standard deviation of the error remained below 5 pixels for all targets considered.

The mean and standard deviation of the measurement error in locating body parts were calculated in pixels. In detail, as is possible to observe in Figure 11, the mean error was lower than one pixel, so the error can be considered zero-centered. Figure 12 shows the standard deviations in horizontal and vertical directions. Although very similar, the standard deviation for the horizontal direction is slightly higher. This was most likely due to the higher speed of the targets in that direction in the frames.

Table 2 shows the mean error and standard deviation in locating body parts averaged among all targets. The results are provided in mm and demonstrate a performance more then adequate for the application. The average error is negligible, demonstrating a casual zero-centered distribution, and the standard deviation is small considering the amplitudes of the displacements involved in the problem. This distribution suggests that light data filtering could be applied to clean the data and make them smoother without significant loss of information, while also taking into account that the targets’ movements are quite smooth. Finally, the method was applied to videos available in the public SVW archive. As there is no information about the cameras used, nor there is no information about the shooting distance, the athlete’s joints trajectory results can only be expressed in pixels. In this case, the average percentage of unrecognized targets rose to 5%, and the average error and standard deviation were around 5 and 10 pixels, respectively.

## 4. Conclusions

In this work, a novel method for the improvement of marker-less pose estimation exploiting FCN was presented. The developed technique is compatible with all the swimming styles, and it is easy to replicate using a single camera positioned below the free surface of the water. On the one hand, the video recording conditions are advantageous for a wide AFOV and for the capture of many swimming cycles, but on the other hand, auto-occlusion and swap among left and right targets during the pose prediction occur. Thereby, an algorithm was developed in order to correctly distinguish mislabeled and swapped left and right body parts. The presented approach (SwimmerNET) was tested on three new videos (i.e., not used for model training) recorded in different pools, with different lighting conditions and targeting athletes with different genders and physiques. The analysis of these videos, regarding locating target body parts and estimating the athlete’s pose, provided a mean error and a standard deviation of approximately 1 and 10 mm, respectively. Benefiting from an accurate and robust measurement of body parts trajectories is of great value for trainers. Indeed, they can quantitatively assess the technical characteristics of each athlete: strength, swimming style and how both influence the athlete’s swimming performance. Moreover, once associated with the anthropometric and ergonometric measurements of the athlete, the performance metrics can really pave the way to a training session specifically tweaked for each swimmer. The method was applied to the videos available in the public SVW archive, resulting in an average portion of unrecognised targets of 5%, and an average error and standard deviation values of approximately 5 and 10 pixels, respectively. Although good results were achieved in terms of recognition of target body parts (i.e., IoU score = 0.999 for all models), this could be improved by increasing the size of the training dataset and diversifying it. The performance level reported here was only achieved through the use of heavy architectures (i.e., the models generated currently weigh about 1 Gbyte each), making it possible to process one frame every 0.8 s. For the purpose of this work, this is not a limitation because this architecture is used by coaches after swimming sessions, and therefore, real-time performance is not required. In future developments, however, through an important increase in the dataset’s size, one could try to train semantic segmentation architectures with lighter and faster backbones (e.g., Mobilenet), thereby paving the way to real-time applications.

## Figures and Tables

**Figure 1 sensors-23-02364-f001:**
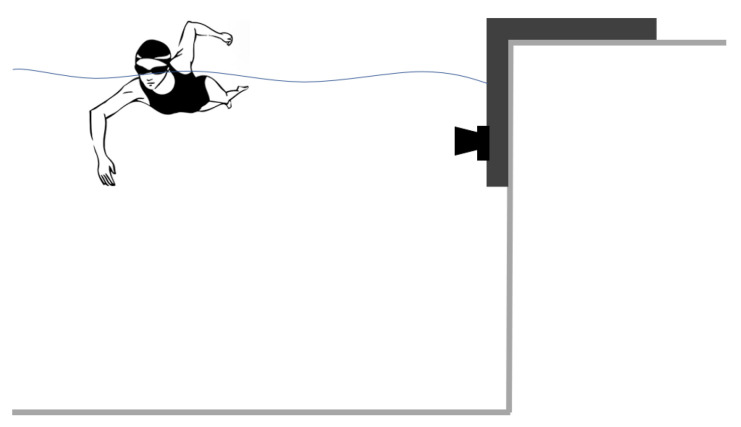
A single camera was fixed sideways underwater, and only the submerged athlete’s body is framed.

**Figure 2 sensors-23-02364-f002:**
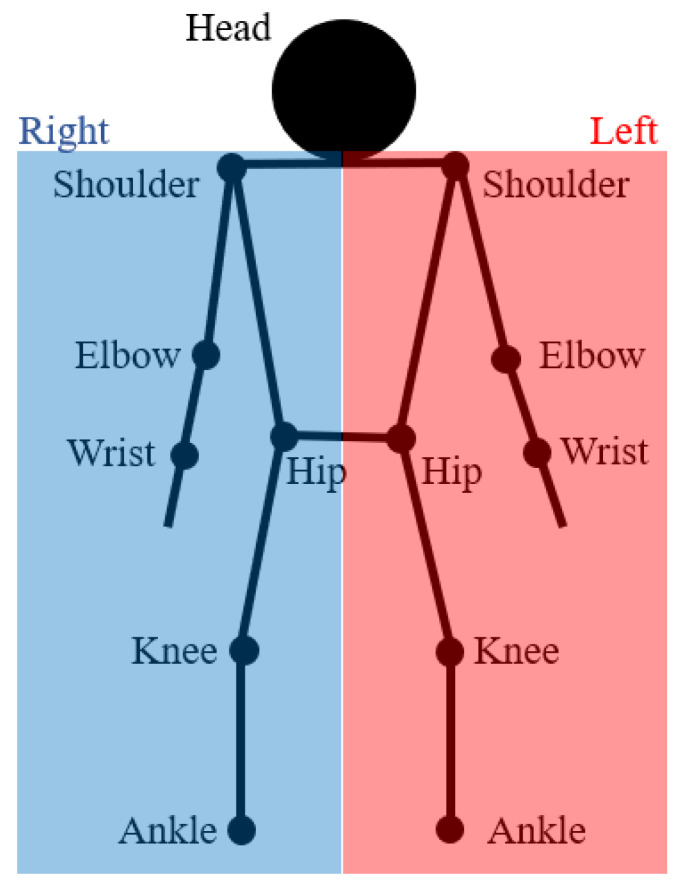
Skeleton that represents a model of a human body. The black dots represent the manually annotated targets.

**Figure 3 sensors-23-02364-f003:**
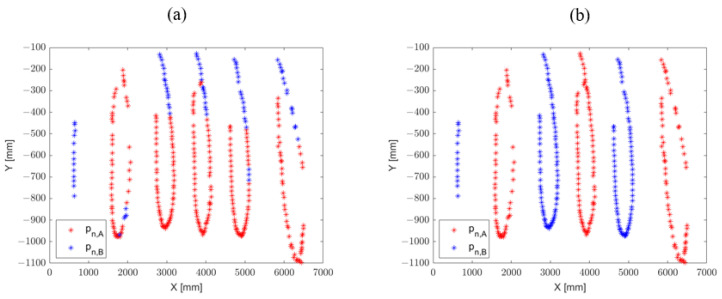
Example of wrist trajectories. (**a**): Curves originally labeled by the models. Labels are highly mixed due to the symmetry of human body with respect to the sagittal plane. (**b**): Curves corrected by the proposed algorithm. In the example, the red curve represents the right wrist’s trajectory and the blue curve represents the left wrist’s trajectory.

**Figure 4 sensors-23-02364-f004:**
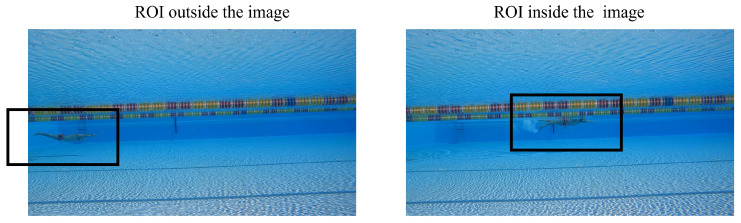
The athlete is identified by the model and used to define a fixed-size ROI of 1024 × 576 pixel in order to obtain small images with the swimmer always in the center of the frame. If the identified ROI falls outside of the input image, the frame is discarded and the next one is used.

**Figure 5 sensors-23-02364-f005:**
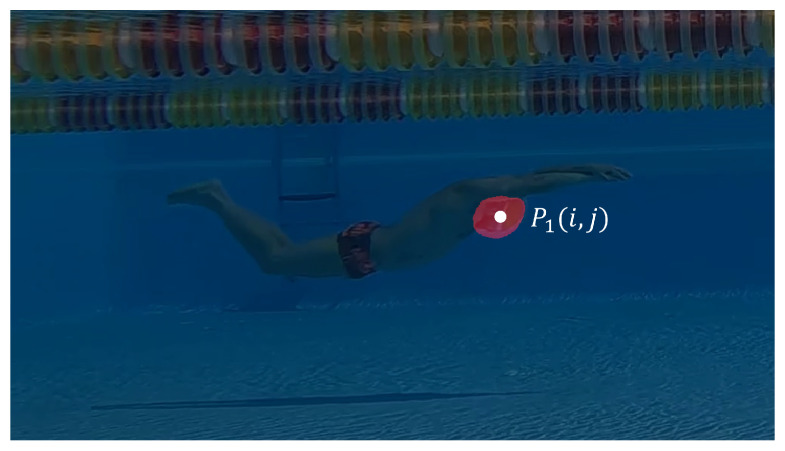
The binary mask identified by the semantic segmentation model for target 1 (i.e., the swimmer’s head) is superimposed on the input image. The position of the target, in pixel, is taken as the position of the center of gravity from the area identified by the model.

**Figure 6 sensors-23-02364-f006:**
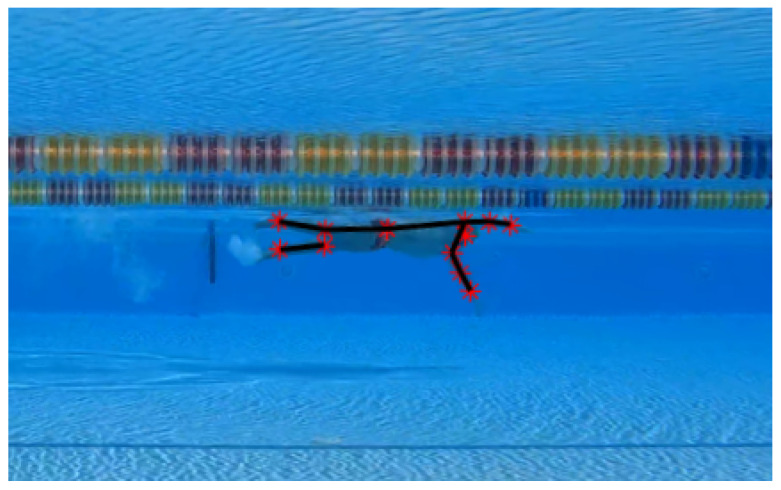
Through iterative application of the developed semantic segmentation models, a coordinate is assigned to each targeted body part that is visible within the frame.

**Figure 7 sensors-23-02364-f007:**
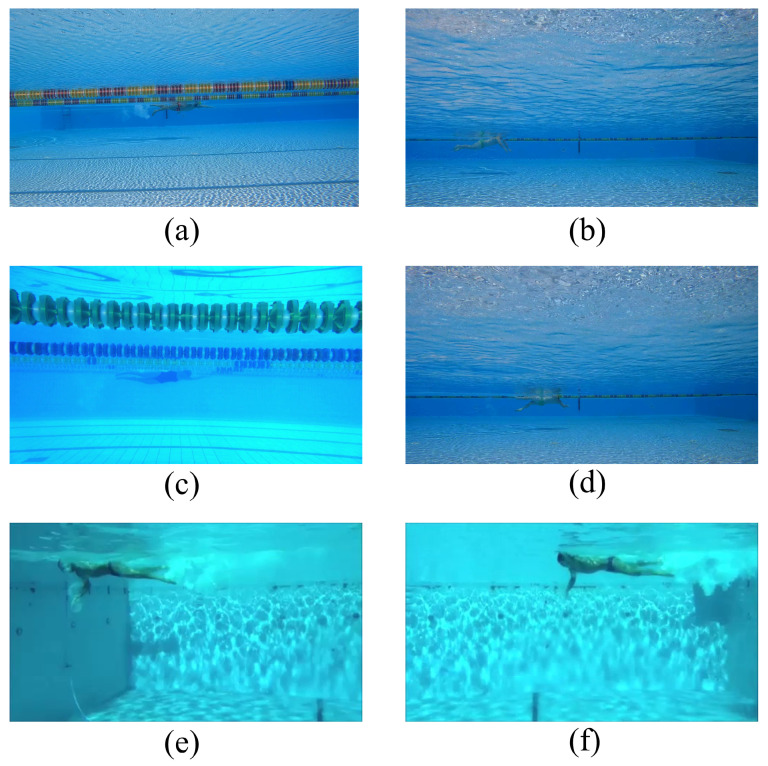
Examples of the different videos used for testing the SwimmerNET method: one athlete swimming freestyle for training (**a**) and two new athletes in different pools for the test phase—specifically, a female athlete performing freestyle (**c**) and a male athlete performing dolphin (**b**) and backstroke (**d**). Finally, there are frames extrapolated from videos gathered from the public Sports Videos in the Wild (SVW) repository [27] showing a male athlete during a freestyle swimming session (**e**,**f**).

**Figure 8 sensors-23-02364-f008:**
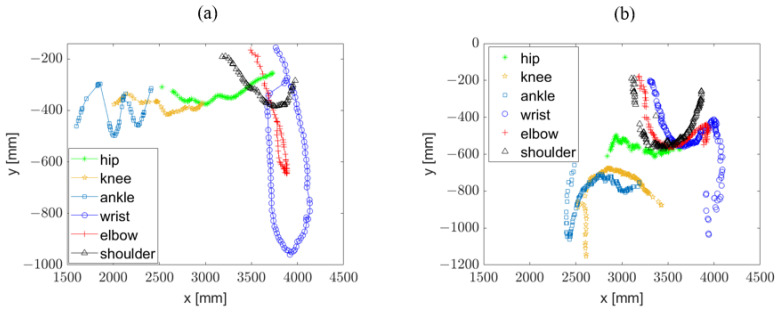
Examples of trajectories of the right parts of the body during a swimming stroke: a 1.90 m tall male athlete during freestyle (**a**) and a 1.80 m tall male athlete during dolphin style (**b**).

**Figure 9 sensors-23-02364-f009:**
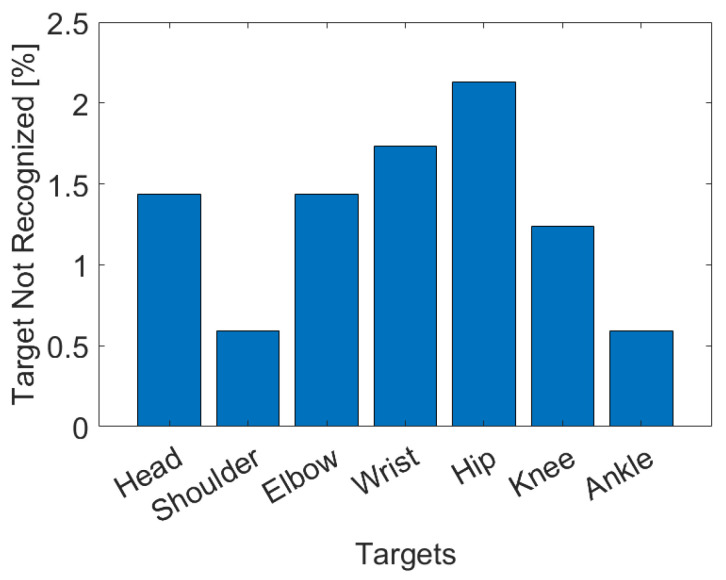
Percentage of targets not recognized by the proposed method divided by body part.

**Figure 10 sensors-23-02364-f010:**
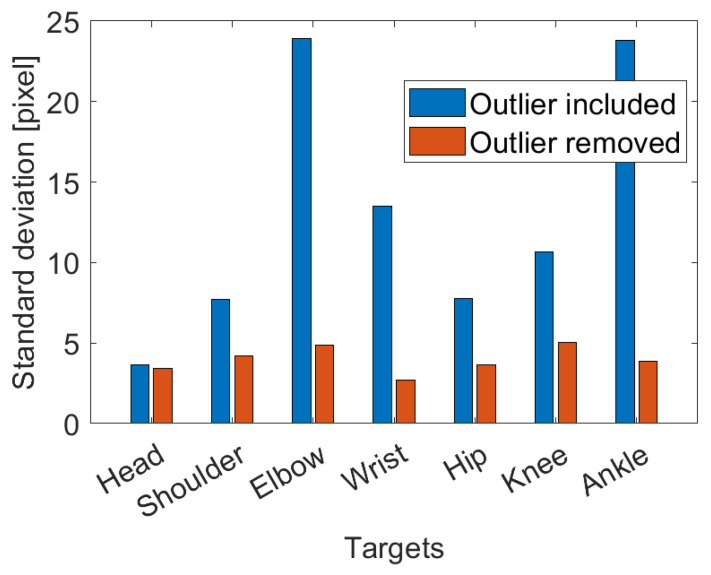
The presence of outliers in the target location increases the standard deviation of the error. Once eliminated, the standard deviation remains below 5 pixels for each target.

**Figure 11 sensors-23-02364-f011:**
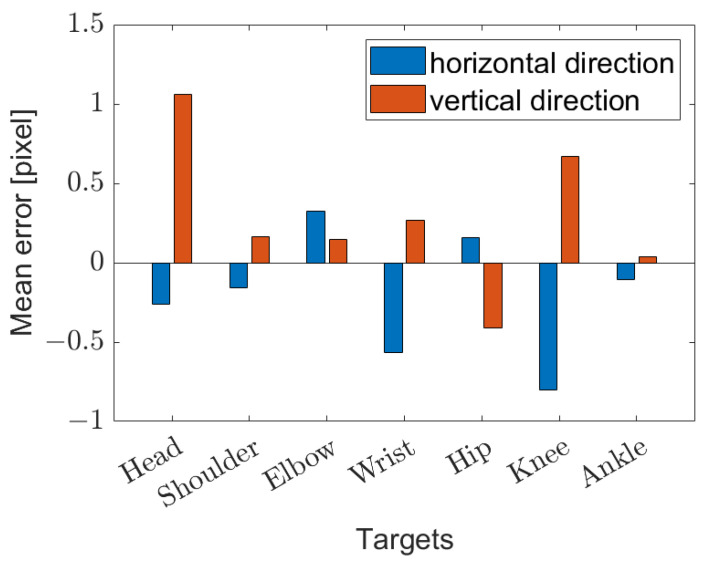
Mean error in locating body parts.

**Figure 12 sensors-23-02364-f012:**
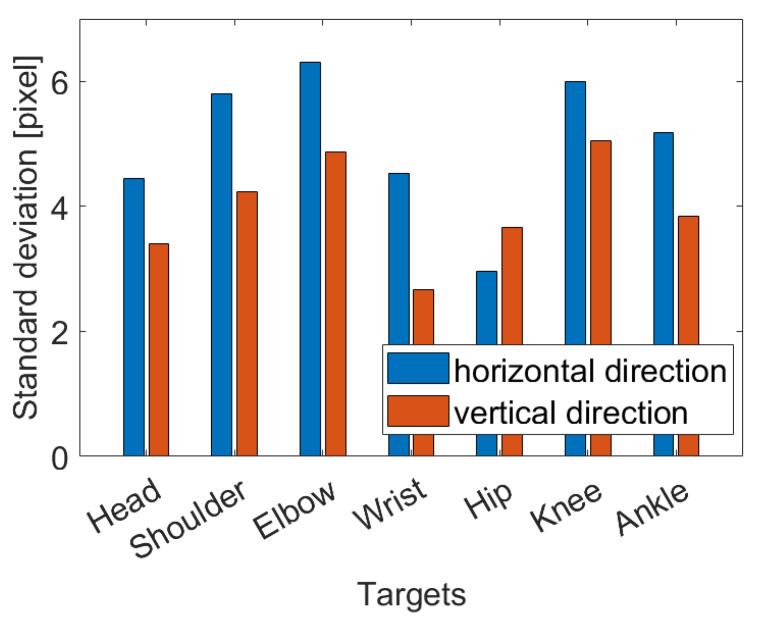
Standard deviation of mean error in locating body parts.

**Table 1 sensors-23-02364-t001:** Models’ performances in identifying diverse targets were trained by optimizing hyper-parameters through a Bayesian approach so as to maximize IoU score.

Target Name	bs	lr
Whole Body	5	0.0031
Head	5	0.0001
Shoulder	8	0.0001
Elbow	8	0.0001
Wrist	8	0.0002
Hip	7	0.0002
Knee	8	0.0015
Ankle	8	0.0011

**Table 2 sensors-23-02364-t002:** Mean error and standard deviation in locating body parts averaged among all targets.

	Mean Error		Standard Deviation	
	[pixel]	[mm]	[pixel]	[mm]
Horizontal direction	−0.2	−0.86	2.6	10.8
Vertical direction	0.3	1.17	2.0	8.6

## Data Availability

The data are not publicly available due to privacy restrictions.

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
