# Peer review of "SwimmerNET: Underwater 2D Swimmer Pose Estimation Exploiting Fully Convolutional Neural Networks"

_sensors, 2023, doi:10.3390/s23042364_

Round 1
Reviewer 1 Report
1. My major concern is regarding why not considering an end-to-end solution, for example, T. Zhou, Y. Yang and W. Wang, "Differentiable Multi-Granularity Human Parsing," in IEEE Transactions on Pattern Analysis and Machine Intelligence, doi: 10.1109/TPAMI.2023.3239194. The work should be discussed in Sec. I.
2. It is not clear to me how the body part segmentation model is trained.
3. It would be essential to evaluate the model performance on popular datasets like LIP, and compare with state-of-the-art methods like Hierarchical human semantic parsing with comprehensive part-relation modeling.
4. The limitations of the proposed method should be discussed.
Reviewer 2 Report
As a general consideration, the study is very interesting as it can be applied to other sports and fields as well.
The authors should mention and comment on a couple of things.
1. First, in general, the authors should mention how long it takes the net to be trained for optimal performance to provide suggestions to the athlete. Just an order of magnitude.
2. Secondly, the authors should explain how they select examples to train the network in case of supervised networks. Do all top athletes have similar movements while swimming or not? So, how to certify optimal movements? Please comment
3. Some point comments:
- the explicit algorithm 1 can be omitted in the paper
- formulas 3 and 4 can be compacted using Vij(f) and aij(f)with i,j belonging to {A,B}
- also figure 4 of the workflow can be omitted as it is explained in the text
- in Table 1 the columns "IoU", "o" and "bb" should be removed as they are constant and do not provide any information
Reviewer 3 Report
This manuscript reported a novel method for the improvement of marker-less poses estimation exploiting FCN was presented. The developed technique is compatible with all the swim styles and it is easy to replicate using a single camera positioned below the free surface of the water. In addition,they also developed an algorithm to correctly distinguish mislabelled and mixed left and right body parts. The presented SwimmerNET approach is tested on three new recorded videos (i.e., not used for model training) providing a mean error and a standard deviation, in locating body target parts and estimating the athlete’s pose, of approximately 1 mm and 10 mm respectively.
The description of this manuscript is detailed and the clear, and the authors have also proved the effectiveness of their algorithm through many experiments. However, the reviewers still have the following questions:
1: The reported algorithm is not innovative enough in terms of network model, but only a simple application of UNet in the field of underwater human pose estimation. In addition, the authors do not give the detailed structure and parameter settings of the network. The reviewers suggest the authors to give the detailed structure and parameter settings of the UNet in this manuscript.
2: Insufficient explanation of why UNet is used. There exist many network models with better performance than UNet in the field of image segmentation and in the field of human pose estimation. The reviewers suggest the authors to analyze the reasons for using UNet in detail or to conduct performance comparison experiments with other network models.
3: The reviewers believe that the experiment is not sufficient. First, the authors did not compare their proposed algorithm with other underwater human pose estimation algorithms. Second, the reviewers felt that testing only on three new underwater videos was insufficient to illustrate the generalization performance of the algorithm.
4: The reviewers did not find the running speed of the proposed algorithm in the manuscript. The reviewers believe that the running speed of the algorithm is very important for underwater human pose estimation.
Based on the above analysis, the reviewers believe that this paper has much room for improvement, so the reviewers recommend a major revision.
Round 2
Reviewer 1 Report
The revision has addressed my concerns.
Reviewer 3 Report
The revised manuscript addresses some of the issues raised by the reviewers, but there are still some major issues that have not been addressed.
1:The reviewers believe that the experiment is not sufficient. First, the authors did not compare their proposed algorithm with other underwater human pose estimation algorithms. Second, the reviewers felt that testing only on three new underwater videos was insufficient to illustrate the generalization performance of the algorithm.
2:The reported algorithm is not innovative enough in terms of network model, but only a simple application of deep neural networks in the field of underwater human pose estimation.
Based on the above analysis, the reviewers believe that this paper still has room for improvement, so the reviewers recommend a minor revision.
